# Dehydroepiandrostenedione sulphate (DHEAS) levels predict high risk of rheumatoid arthritis (RA) in subclinical hypothyroidism

**Ravindra Shukla** [1] *, **Mayank Ganeshani** [1,2], **Monica Agarwal** [1], **Rakesh Jangir** [1,3], **Gaurav Kandel** [1], **Shrimanjunath Sankanagoudar** [4], **Shival Srivastava** [5]

1 Department of Endocrinology, AIIMS, Jodhpur, India, 2 Department of Orthopaedics, SMS, Jaipur, India, 3 Department of Paediatrics, BKMC, Bikaner, India, 4 Department of Biochemistry, AIIMS, Jodhpur, India, 5 Department of Physiology, AIIMS, Jodhpur, India

* ravindrashukla2@rediffmail.com, shuklar@aiimsjodhpur.edu.in

## Abstract

### Introduction

The presence of rheumatism is well recognized in primary hypothyroidism. Dehydroepiandrstenedione sulphate (DHEAS) is associated with rheumatological diseases like rheumatoid arthritis (RA) and systemic lupus erythematosus (SLE). This study aims to explore relationship between joint pains and DHEAS levels in primary hypothyroidism.

### Methods

Retrospective study of 78 subjects with subclinical hypothyroidism, with TSH within reference range. The joint pains were evaluated by European Union League against rheumatism (EULAR-CSA) score and compared with serum DHEAS, RA factor, Anti-TPO antibody, highly sensitive C-recative protein (hsCRP), vitamin D levels.

### Result

DHEAS levels <43.6 mcg/dl significantly predicted clinical features of pre RA as assessed by EULAR CSA criteria with acceptable specificity (82%). EULAR CSA score is fairly valid in assessing imminent RA in primary hypothyroidism.

### Conclusion

Lower DHEAS predicts clinical features of imminent RA in subjects with primary hypothyroidism. This is akin to low DHEAS seen in many rheumatological disease with possibly similar mechanism. Another possibility is low DHEAS alters hepato-hypothalamo pituitary adrenal axis in presense of cytokines and induces a hitherto unrecognized state of pre rheumatoid arthritis like syndrome. Future studies on primary hypothyroidism should focus on role of lower DHEAS levels in inducing symptoms of fatigue and joint pains.

**Data Availability Statement:** The data that support finding of study is available in excel format at the Open Science framework (https://osf.io/t4gxj/?

view_only=
614b6b07b0c7465ab50d1b06b5b4878f) under
project name "Clinico-biochemical profile of
primary hypothyroidism in Western Rajasthan."

**Funding:** The authors received no specific funding
for this work.

**Competing interests:** The authors have declared
that no competing interests exist.

**Abbreviations:** RA, Rheumatoid Arthritis; EULAR,
European league against Rheumatism; CSA,
Clinically Significant Arthralgias; Anti-CCP, Anti
cyclic citrullinated peptide; DHEAS,
Dehydroepiandrostenedione Sulphate; SLE,
Systemic Lupus Erythematous; Anti-TPO Ab, Anti-
thyroperoxidase Antibody; hsCRP, high sensitive
C- reactive protein; TSH, Thyroid Stimulating
Hormone; AUC, Area under curve; T3, tri-
iodothyronine; T4, tetra-iodothyronine; rT3, reverse
tri-iodothyronine; QoL, Quality of Life; IL-6,
Interleukin 6; IL-10, Interleukin 10; IL-4, Interleukin
4; HPA, Hypothalamopituitary Axis; 11βHSD1, 11β
hydroxysteroid dehydrogenase-1.

# Introduction

Rheumatism is well recognized symptom in primary hypothyroidism While rheumatoid
arthritis (RA) is associated with primary hypothyroidism, large subset of patients do not fulfill
criteria for RA [1] European Union League against Rheumatism (EULAR) has released a vali-
dated criteria for clinically suspect arthralgia (CSA) to determine early RA. The EULAR-CSA
criteria makes it possible to identify high risk of RA on clinical examination without need for
RA factor or anti CCP antibodies [2]. Dihydroepiandrostenedione sulfate (DHEAS) is most
abundant adrenal steroid, immunomodulator and found to be low in autoimmune diseases
affecting joints e.g. RA, SLE [3] Low DHEAS has been documented previously in primary
hypothyroidism [4]. We hypothesized DHEAS to be associated with arthralgias in autoim-
mune primary hypothyroidism. The study objective was to examine association of EULAR
CSA criteria with DHEAS levels in primary hypothyroidism patients on adequate levothyrox-
ine dose.

# Subjects & methods

Retrospective data on primary hypothyroidism has been collected under the project "Clinico-
biochemical profile of primary hypothyroidism in Western Rajasthan" between 2016–2018 in
Department of Endocrinology AIIMS, Jodhpur. The study was approved by Institute Ethics
Committee vide AIIMS/IEC/2017/357. The retrospective data was fully anonymized before
access. The electronic medical records in the hospital are identified with unique bar code.
Even the bar code patient ID was removed during statistical analysis. The IEC waived consent
for retrospective data collection. Inclusion criteria used was: a diagnosis of primary hypothy-
roidism, Thyroid Stimulating Hormone (TSH) levels 2-5mU/L, at least one DHEAS value
within 1 week of TSH, CSA evaluated as per EULAR consensus criteria, on levothyroxine
replacement (between 25–125 mcg). Those with pregnancy, RA, SLE, diabetes, cardiovascular
event, adrenal disease, or any other chronic illness were excluded. The sample size was calcu-
lated with 80% power to detect a difference of 20 mcg/dl serum DHEAS in both ways (bi-
directional significance at p <0.05) with respect to EULAR score. The following formula was
used, which gave sample size of 80 [5]:

$n = 2\ (Z_{\alpha}+Z_{1-\beta})^2\ \sigma^2 \div d^2$ where

$Z_{\alpha}$ = 1.96 for 2 tailed results at p < 0.05

$Z_{1-\beta}$ = 0.8416 for power of 80%

σ = Standard Deviation = 50 mcg/dl calculated on based on previous study [6]

d = effect size = 20 mcg/dl

A total of 78 patients record were included out of 462 (Fig 1) and values for DHEAS, 25
hydroxyvitamin D, Anti TPO, Rheumatoid Arthritis (RA) factor, highly sensitive C-reactive
protein (hsCRP) was obtained. Automated chemilumiscence was used to measure TSH (Advia
Centaur XP,) DHEAS (Diasorin), 25 (OH) vitamin D (Advia Centaur XP), Anti TPO antibody
(Advia Centaur XP). Rheumatoid factor was measured by immunoturbidometry method
(Beckman-Coulter). A value of > 18 IU/ml and >16 IU/ml was considered positive for Anti-
TPO and RA factor, respectively. The inter-assay and intra-assay Co-efficient of Variation
(C.V.) was <10% for all parameters measured.

EULAR in its position statement [2] had released a list of seven objective clinical param-
eters for diagnosis of clinically suspicious arthralgia (CSA) [henceforth called EULAR-CSA
score] likely to progress to rheumatoid arthritis. EULAR CSA score has been incorporated
in clinical practice of evaluating features of imminent RA in our Thyroid Clinic. Since this
was a retrospective study, assessors of EULAR score were not aware of DHEAS levels of
respective subjects. A positivity of > 4 is likely to have increased risk for rheumatoid

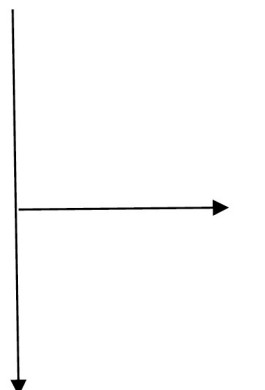

**Fig 1. Study flow chart.** Of total 462 datasets screened, 78 datasets were included in present study.

arthritis [2]. EULAR- CSA score > 4 also predicted arthritis as evidenced by MRI. Hence study subjects were divided into two groups—one with score less than or equal to 3 (hence forth called CSA absent) and another with score more than or equal to 4 for analysis (henceforth called CSA present).

## Statistical analysis

The age and DHEAS levels were log transformed. A bivariate correlation was performed between log age and log DHEAS. EULAR criteria was outcome variable. Normality of data was assessed using Kolmogarov-Smirnov test. Multiple linear regression was then performed to analyze whether DHEAS predicts joint pains as assessed by EULAR criteria with predictive variables being age and DHEAS. One way ANOVA was used separately with each of the variables (age, DHEAS, Anti TPO Ab, RA factor, vitamin D, hs CRP) to determine whether they are associated with EULAR CSA score. A correlation analysis of EULAR score with each of the above variable was also performed. Pearson correlation was used for parametric (age, DHEAS), Spearman correlation for non-parametric (RA factor, hsCRP) while Kendal Tau was used for non-parametric dependent Anti TPOAb. Only absolute values of anti TPO antibody and RA factor were taken in analysis. The DHEAS levels were divided into quartiles (supplementary material in S1 File). Receiver operator curves (ROC) were performed with each quartile to determine which DHEAS quartile predicts rheumatism as assessed by EULAR–CSA score. A binary logistic regression analyses was performed to ascertain the effect of DHEAS in first quartile on likelihood that subjects have on individual component of EULAR-CSA score and whether they have CSA score >4. A p value <0.05 was considered significant. SPSS version 21 was used for statistical calculation.

**Table 1. Descriptive characteristics of study subjects.**

|  | Number (n) | Mean(+/- S.D.) | Skewness Static |
|---|---|---|---|
| Age (in years) | 78 | 43.23 +/- 10.21 | -0.2 |
| TSH (mU/L) | 78 | 3.06 +/- 3.03 | 8.3 |
| DHEAS (mcg/ml) | 78 | 99.30 +/- 88.28 | 1.9 |
| Vitamin D (ng/ml) | 14 | 75.41 +/- 177.90 | 3.4 |
| Anti -TPO Ab (IU/L) | 8 | 352.02 +/- 321.47 | 0.05 |
| hs CRP (mg/dl) | 12 | 11.99 +/- 22.90 | 2.8 |
| RA factor | 16 | 15.64+/-30.33 | 3.6 |

As shown above Age, DHEAS, Anti TPO antibody were normally distributed (skewness static <2).

## Results

All eligible patients were females. Mean TSH was 2.8 mU/L. Vitamin D, Anti-TPO Ab, hs CRP and RA factor was not available for all the subjects (Table 1). Age, DHEAS, Anti TPO antibody were normally distributed (Table 1). DHEAS level was found to be inversely correlated with age in years (r = 0.241, p = 0.03) (Fig 2).The correlation was weak (spearman rho coefficient of correlation = -0.241) but statistically significant (p = 0.03).There was no statistically significant correlation of Anti TPO Ab (r = -0.12, p = 0.69), vitamin D (r = 0.11, p = 0.61), RA factor (r = 0.4, p = 0.051), hsCRP (r = 0.46, p = 0.53) with EULAR-CSA (Table 2) On multiple regression analysis, serum DHEAS predicted rheumatism as defined by EULAR CSA score, F

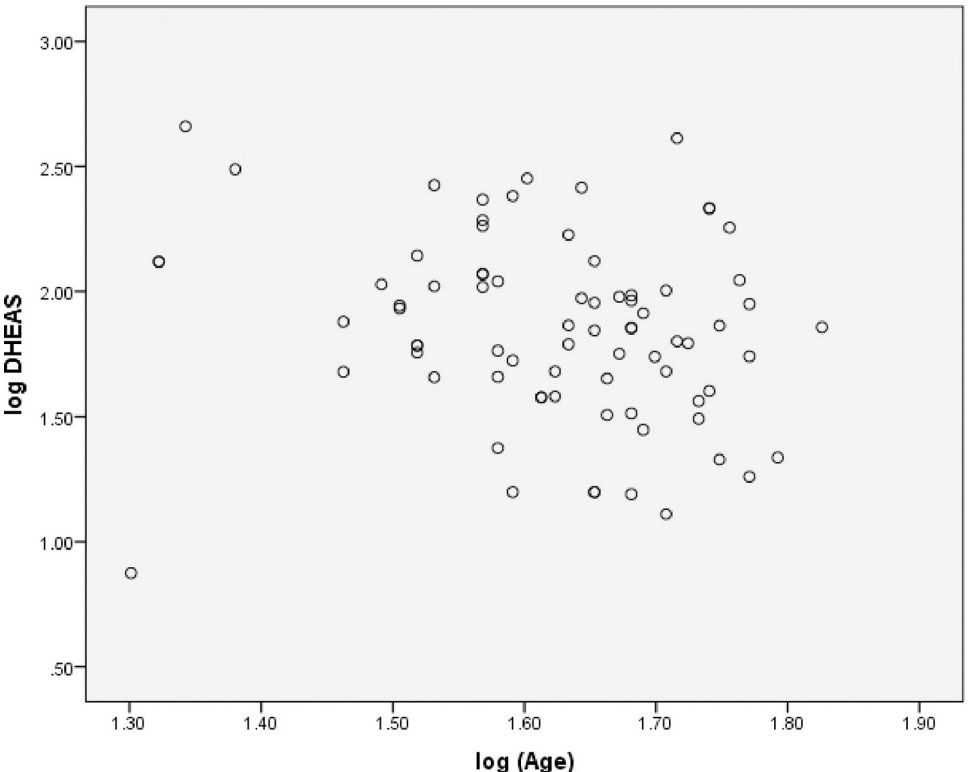

**Fig 2. Inverse correlation between age and DHEAS levels.** The correlation was weak (spearman rho coefficient of correlation = -0.241) but statistically significant (0.03).

**Table 2. Result of correlation of different parameters with EULAR-CSA score.**

|  | Bivariate correlation co-efficient | p-value |
| --- | --- | --- |
| Age | 0.048 | 0.64 |
| DHEAS | -0.224 | 0.049 |
| Anti-TPO | -0.12 | 0.69 |
| 25(OH) Vitamin D | 0.11 | 0.61 |
| Hs CRP | 0.46 | 0.53 |
| RA factor | 0.49 | 0.051 |

correlation with DHEAS.

DHEAS was inversely correlated with EULAR-CSA score and was statistically significant. RA factor was positively correlated but statistically insignificant.

(1,11.5) = 4.09, p = 0.049. Only the ROC curve for the first DHEAS quartile predicted joint pains in primary hypothyroidism (Fig 3). The AUC was 0.67 (C.I = 0.52–0.83, p = 0.021) (Fig 3). The first quartile serum DHEAS value was <43.6 mcg/dl. Thus, DHEAS less than 43.8

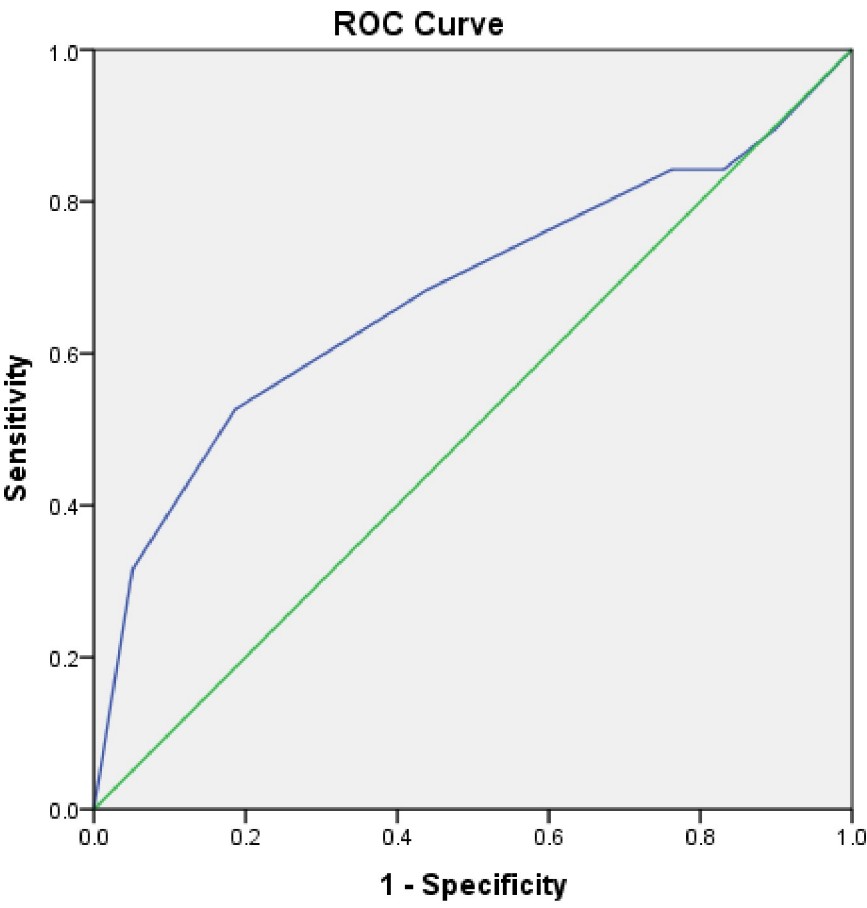

Diagonal segments are produced by ties.

**Fig 3. Low DHEAS as a predictor of rheumatism in primary hypothyroidism.** ROC curve for low DHEAS as a predictor of rheumatism in primary hypothyroidism. The AUC curve was 0.67 (C.I = 0.52–0.83, p = 0.021).The cut off point for <43.6 mcg/dl was found. DHEAS, 43.8 predicted CSA of >5 with 82% specificity but low sensitivity (52%).

**Table 3. Results binary logistic regression analyses to ascertain the effect of DHEAS on different variables.**

|  | OR (95% C.I.) | P value |
|---|---|---|
| CSA present | 1.01 (1–1.01) | 0.181 |
| Onset < 1 year | 1.01 (1–1.01) | 0.6 |
| **MCP joint involvement** | 1.01 (1–1.01) | **0.044** |
| Morning stiffness > 1 hour | 1.01 (1–1.01) | 0.37 |
| Symptoms more in morning | 1.01 (1–1.01) | 0.38 |
| First degree relaive with RA | 1.01 (1–1.01) | 0.63 |
| Difficulty in making fist | 1.01 (1–1.01) | 0.062 |
| Positive squeeze test | 1.01 (1–1.01) | 0.13 |

Only MCP joint involvement was found to be statistically significant. Low DHEAS does not increase likelihood of having CSA. The results affirm symmetry of EULAR CSA score in evaluating primary hypothyroidism in context of decreased DHEAS.

mcg/dl predicted CSA of >5 with 82% specificity and 52% sensitivity. Although serum DHEAS < 43.6 was not a very strong predictor of presence of joint pains in primary hypothyroidism (mean AUC = 0.67), it was statistically significant. Binary logistic regression analysis revealed that DHEAS in lowest quartile (< 43.6 mcg/dl) predicted variables of EULAR CSA score (Table 3). Low DHEAS predicted metacarpophalangeal joint involvement and was found to be statistically significant (p = 0.044). It correctly classified 91% of cases but explained

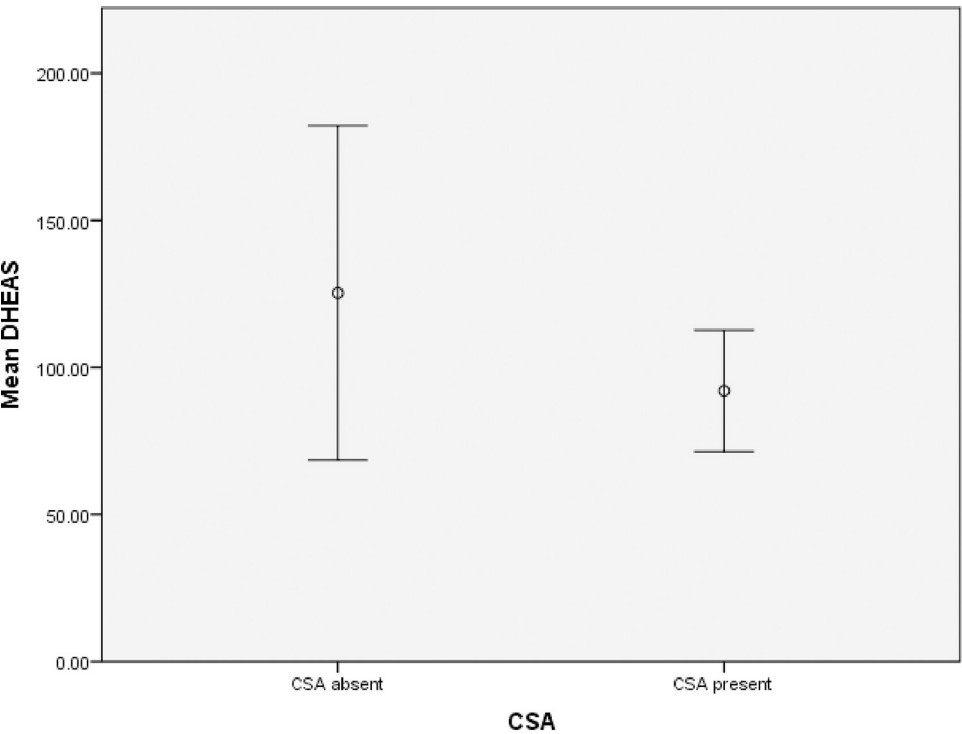

**Fig 4. Comparison of mean DHEAS according to CSA status.** CSA was considered present if CSA score was greater or equal to 4. The mean DHEAS of those with CSA present (125.29 +/- 26.81) was lower than those with CSA absent (80.60 = /- 10). However it was not statistically significant. (p = 0.17).

only 10% variance (Nagelkerke $R^2$ = 0.12), suggesting a limited clinical value. The Odds Ration (OR) was around 1 for all seven components of EULAR-CSA score (Table 3), suggesting no single criteria can be used in isolation while evaluating joint pains in primary hypothyroidism. The results affirm symmetry of EULAR CSA score in evaluating primary hypothyroidism in context of decreased DHEAS. The mean DHEAS of those with CSA present (125.29 +/- 26.81) was lower than those with CSA absent (80.60 = /- 10). However, it was not statistically significant (p = 0.17). Despite low DHEAS predicting high grade of arthritis, mean DHEAS levels in those with EULAR CSA score > 4 were not statistically significant from those with lower CSA values (Fig 4).

## Discussion

### Decreased QoL and non-specific symptoms in subclinical primary hypothyroidism

Primary hypothyroidism is most common organ specific autoimmune disease. The treatment for primary hypothyroidism involves thyroid hormone replacement, mostly in form of oral levothyroxine [7]. Therapy is monitored by periodic TSH levels and clinical evaluation. Despite adequate levothyroxine supplementation as assessed by thyroid hormone levels, pain and impaired QoL persist [8]. These non specific symptoms are attributed to other associated disorders, somatization or inadequate peripheral 3,3',5-Triiodothyronine (T3) availability. Peripheral T3 is availability is dependent on T4 to T3 conversion and also degradation of T4 & t3 to inactive 3,3',5' Triiodothyronine (rT3). This in turn is dependent on ubiquitous enzymes -the deiodinases [9]. There has been much interest in exploring therapeutic avenues to ameliorate non specific symptoms and improve this "localised" hypothyroidism [10]. Localized hypothyroidism of synovial cells have been documented in euthyroid subjects of RA to the extent that fT3 is nearly absent in synovial fluid [11]. Just like primary hypothyroidism, impaired QoL persist despite controlled disease activity in rheumatological diseases like RA, systemic lupus erythemtosus (SLE), Sjogren syndrome [12]. Also, autoimmune hypothyroidism is strongly associated with disease like RA, SLE, Sjogren syndrome leading to overlap of symptoms [13].

### DHEAS and autoimmunity

DHEAS decrease with age and are also low in premenopausal females [14]. We found DHEAS to be inversely related with age. Physiologic conditions with high DHEAS levels have lower incidence of rheumatological diseases e.g. males, post menopausal females. DHEAS is important adrenal androgen in context of rheumatological diseases. Previous studies have documented low DHEAS levels in SLE, RA and Sjogren syndrome [15,16]. Here, we find DHEAS levels to be inversely correlated with EULAR-CSA score in females with primary hypothyroidism. The relationship between DHEAS levels and arthralgia in primary hypothyroidism can be explained by three hypotheses.

### Cytokine hypothesis

The primary hypothyroidism is associated with increase in serum cytokines like Interleukin-6 (IL-6), tumor necrosis factor alfa (TNF-α) and hs CRP [17] and possibly IL 10, IL 4 [17]. Although levothyroxine brings down the level of cytokines, they do not return to same level as healthy controls after euthyroid state is achieved [18]. Paradoxically, T4 (but not T3) may induce IL-6 production in synovial cells [11]. Since IL-6 is primary mediator in pathogenesis of rheumatoid arthritis [19]; this common pathogenetic link explains the usefulness of EULAR-CSA score in hypothyroidism. DHEAS has suppressive effect on inflammatory

cytokines, and local DHEA deficiency precedes synovial joint inflammation [20]. Thus low DHEAS in primary hypothyroidism can lead to clinical features of early RA.

## Bystander association

The decreased DHEAS could very well be a bystander effect of generalized HPA axis suppression in these subjects. Adrenocortical function has been found to be decreased in patients primary hypothyroidism taking levothyroxine therapy [21]. Decrease HPA axis activity has also been postulated as pathogenetic mechanism of RA. The circadian dip in serum cortisol leads to increased cytokine build up early morning, leading to increased morning stiffness [22]. Increased DHEAS but not serum cortisol, androstenedione or testosterone was associated with improved disease activity in RA patients [23]. This points to a role of DHEAS independent of HPA axis, and questions the bystander association.

## Role of 11β hydroxysteroid dehydrogenase-1 (HSD1)

A novel concept of hepato-hypothalamo-pituitary-adrenal-renal axis postulates the important role of 11 β HSD enzymes in HPA axis regulation in context of rheumatological diseases [24]. 11 beta HSD1 has been involved in cortisone conversion to active cortisol at tissue level and thus important for localized immunomodulation. Both thyroid hormone and DHEAS inhibit 11βHSD1 enzyme. 11βHSD1activity has been found to be increased in hypothyroidism [25]. Increased 11βHSD1 activity in pituitary, in presence of inflammatory cytokines, leads to downregulation of HPA axis [24]. This may lead to joint pains only when low DHEAS is co-existent with hypothyroidism.

## DHEAS levels and CSA severity

DHEAS in lowest quartile (<43.5) predicted clinical features of pre-RA. The decreased serum DHEAS does not favorably predict a particular EULAR CSA component over other. This symmetric prediction makes likely the conclusion, that serum DHEAS < 43.5 might induce a pre-RA like syndrome in women with primary hypothyroidism on adequate levothyroxine replacement. TNF-α which is raised in primary hypothyroidism [7] can lead to localized DHEA deficiency in synovial cells [9]. It can be hypothesized that low DHEAS (i.e <43.5 mcg/dl) induces localized androgen deficiency and synovitis, seen in the study subjects. The mean DHEAS level of CSA present group was lower as compared to CSA absent group but was statistically not significant. This in important in several ways. First, subjects in former group had DHEAS levels (80 mcg/dl) much higher than those described in RA subjects (18 mcg/dl) [8] These were not RA patients (see exclusion criteria) and may never progress to overt RA. Rather tissue hypothyroidism may be responsible for arthralgias. Secondly, there exists no cut-off of DHEAS where severity of rheumatism increases in disproportionate manner. There may exist a spectrum of joint pains predicted by DHEAS in primary hypothyroidism patients which clinically follows RA like onset, but its progression is not determined by increased positivity of CSA components. However, testing of the assumption would need a prospective follow up. The result of binary logistic regression implied that all of the EULAR parameters have equal value in terms of weightage. There exists no gold standard to evaluate joint pains in primary hypothyroidism, which could be used to compare EULAR CSA score. But based on symmetry of Odds Ratio (OR), EULAR CSA score can be taken as valid measure to evaluate features of pre RA in primary hypothyroidism for future studies.

## Novel therapeutic approach to autoimmune hypothyroidism

Our findings advocate an approach to primary hypothyroidism from lens of rheumatological disease. DHEAS has been tried in SLE with some success and has been suggested as therapeutic option in PMR [26]. Thus, DHEAS supplementation may be evaluated in hypothyroidism and considered for those with lower serum DHEAS (< 43mcg/dl) and arthralgias. Conversely, T3 +T4 combination which have been used to treat localized hypothyroidism, can be used in hypothyroid subjects with arthralgias and by extension in RA. Secondly, it is likely that our study subjects had low QoL. Although low QoL in primary hypothyroidism is documented, more studies are needed to ascertain role of DHEAS with decreased QoL in primary hypothyroidism. Lowered oxidative defense in overt and sub-clinical hypothyroidism may affect QoL [27] and DHEAS has protective effect during oxidative stress [28] Thirdly, all of the patients fulfilling study criteria were women. Although rheumatological diseases are more common in women, this finding implies to rule out joint pains in women with subclinical hypothyroidism.

This study unravels research questions. Is low DHEAS also associated with complaints of generalized pain and fatigue in patients of primary hypothyroidism? Is intervention with Disease Modifying Anti Rheumatoid Arthritis Drugs (DMARD) like hydroxychloroquine indicated in women of primary hypothyroidism with joint pains? and most important, although DHEAS is well associated with RA onset in prospective study, what is relationship of DHEAS with respect to EULAR CSA criteria in general?

## Study limitations

Our study had certain drawbacks. This was retrospective cross sectional study. Only those with EULAR CSA examination done were included. It is possible that those with low DHEAS were examined more closely for RA-like symptoms and EULAR scoring done. Secondly, we did not measure serum inflammatory cytokines. This would have been crucial in proving cytokine hypothesis. Thirdly, RA was excluded on clinical basis, no anti-cyclic citrullinated peptide (CCP) antibody was done. Finally we analyzed low DHEAS as compared to EULAR CSA which is fairly specific scoring system, but we may have missed joint symptoms not covered EULAR CSA, yet important in context of primary hypothyroidism.

## Conclusion

This study finds low DHEAS levels to predict features of imminent RA in women with primary hypothyroidism. There can be three possible hypotheses to explain the finding. This study emphasizes that those on adequate levothyroxine replacement, as judged by thyroid hormones in reference range, may not be physiologically euthyroid. This study also used EULAR CSA criteria for the first time in hypothyroidism subjects. Future studies on primary hypothyroidism should focus on role of DHEAS in improving low QoL.

## Supporting information

**S1 File. Supplementary material 1: The EULAR-CSA score.**
(DOCX)

## Acknowledgments

The authors acknowledge contribution of Dr Divyangi Mishra in data collection.

## Author Contributions

**Conceptualization:** Ravindra Shukla, Rakesh Jangir.

**Data curation:** Mayank Ganeshani, Monica Agarwal, Shrimanjunath Sankanagoudar.

**Formal analysis:** Monica Agarwal, Shrimanjunath Sankanagoudar.

**Methodology:** Rakesh Jangir.

**Project administration:** Gaurav Kandel.

**Software:** Shival Srivastava.

**Supervision:** Shival Srivastava.

**Validation:** Mayank Ganeshani, Gaurav Kandel.

**Visualization:** Shival Srivastava.

**Writing – original draft:** Ravindra Shukla.

**Writing – review & editing:** Ravindra Shukla.

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
