## [Decision Letter · Decision Letter 0]

20 Nov 2020

PONE-D-20-22641

Dehydroepiandrostenedione Sulphate  (DHEAS)  Levels Predict high risk of  Rheumatoid Arthritis (RA) in Subclinical Hypothyroidism

PLOS ONE

Dear Dr. 

Thank you for submitting your manuscript to PLOS ONE. After careful consideration, we feel that it has merit but does not fully meet PLOS ONE’s publication criteria as it currently stands. Therefore, we invite you to submit a revised version of the manuscript that addresses the points raised during the review process.

Please submit your revised manuscript by 15 days If you will need more time than this to complete your revisions, please reply to this message or contact the journal office at plosone@plos.org. Please include the following items when submitting your revised manuscript:

We look forward to receiving your revised manuscript.

Kind regards,

Rosanna Di Paola, MD

Academic Editor

PLOS ONE

Journal Requirements:

Reviewers' comments:

Reviewer's Responses to Questions

**Comments to the Author**

1. Is the manuscript technically sound, and do the data support the conclusions?

Reviewer #1: Partly

Reviewer #2: Partly

2. Has the statistical analysis been performed appropriately and rigorously? 

Reviewer #1: I Don't Know

Reviewer #2: Yes

3. Have the authors made all data underlying the findings in their manuscript fully available?

Reviewer #1: Yes

Reviewer #2: Yes

4. Is the manuscript presented in an intelligible fashion and written in standard English?

Reviewer #1: No

Reviewer #2: Yes

5. Review Comments to the Author

Reviewer #1: Several abbreviations are used in the abstract without defining them: DHEAS, EULAR CSA, TPO, hsCRP.

While the meaning is clear and understandable, the manuscript should be edited for English grammar.

The use of hsCRP rather than CRP for the measurement of inflammation in the context of arthritis is not universally accepted, as the meaning of hsCRP in this context is not as well-documented as that of CRP or ESR.

If ESR data are available, these should also be included.

The absence of males from the cohort significantly weakens the generalizability of the results.

The sensitivity of DHEAS for predicting CSA was low, DHEAS levels < 43.6 while statistically significant did not have very strong predictive value of joint pains, and as seen in Figure 4, the confidence intervals of the mean DHEAS levels in those who did and did not meet CSA status were overlapping.

The statement in the discussion that DHEA supplementation has a proven role in SLE and PMR is not widely accepted, and the authors provide only one reference article to support this controversial statement in SLE and no evidence at all to support such a claim in PMR.

Overall, the paper's conclusion that the potential role of DHEAS in improving low QoL in hypothyroid patients with joint pain and normal TSH should be further investigated is not strongly supported by the findings described.

Reviewer #2: To start, the terminology, methods, and hypothesis used in this study are confusing. Is this a study to demonstrate a correlation of DHEAS levels with “early RA,” pre-RA, or simply with “rheumatism” - which I take to mean joint pains and arthralgia but not clinical synovitis (the term “rheumatism” is not precisely defined by the authors). This is not a trivial point. In fact, it is central to the whole narrative of the study. Yet, all of these terms are used interchangeably throughout the paper.

To be clear, “early RA” is not “pre-RA”, and rheumatism can describe any, all, or neither of those terms. The EULAR-CSA score was established to identify specific patients in the pre-clinical stage of RA disease development, as RA likely develops in a multi-step approach from pre-clinical autoimmunity to overt systemic and articular inflammatory disease. The score does not determine who has early (classifiable) RA. Indeed, the EULAR CSA score was validated against clinical acumen that was able at its best to identify a 20% chance of someone with CSA developing early RA in the next few years.

With that point made, the authors then study whether there is a relationship between DHEAS levels in subclinical hypothyroidism and “rheumatism” or “early RA” as measured by a the CSA score. Here are some issues:

1. DHEAS levels significantly and inversely correlated with CSA score. However – they did NOT correlate with CSA being present. When the CSA score was dichotomized, binary logistic regression found no association (an odds ratio of 1.01) between whether CSA was present or absent and DHEAS levels. And mean DHEAS was not significantly different in those with CSA and those without. So the statement that “serum DHEAS predicted rheumatism” is NOT correct (line 109) based on those data. Again, it depends upon how one defines “rheumatism,” because MCP pain can be considered a feature of “rheumatism,” but DHEA as a whole did not correlate with higher risk arthralgias predictive of developing RA as defined by a CSA score>4.

2. When DHEAS levels were dichotomized as well, then there was a correlation between a DHEAS level <43.6 and CSA present. The author’s state that “low levels of DHEAS” predict clinical features of “early RA” in primary hypothyoid patients. This is misleading. First, is DHEAS < 43.6 low or not? Physiologic rather than pathological levels of DHEA may be relevant to the immunologic/inflammatory effects mentioned in the paper. No reference is made to what is considered a normal level physiologically in women other than general mention of lower and higher levels by age and menopausal status. Second, the authors selectively drew their cutoff at that level not because it was low physiologically per se but because it granted them the most favorable characteristics on an ROC curve, such that only those DHEAS levels in the bottom quartile were considered. And even then, the AUC was weak. Third, because DHEAS levels correlate with age and other factors (as shown, albeit weakly) selectively reporting results comparing only the lowest quartile could introduce bias by comparing patients with different demographic characteristics (and lower DHEAS levels) to those with higher levels and different demographics.

Other issues:

3. Were the assessors of CSA blinded to DHEAS levels? I assume since CSA is part of the clinical practice and that this is a retrospective review, the answer is that the assessors had no knowledge of present or past DHEAS levels. But that needs to be stated.

4. How were patients with RA excluded? Was this based on ACR 1987 or ACR/EULAR 2010 criteria? Or simply a “clinical diagnosis” of RA. If the latter, then this is problematic. The authors did not measure CCP levels in their patients, but CCP+ patients (even those without arthritis) could have a diagnosis of “RA” in their chart because of the positive test and thus could have been excluded. If so, then the authors may have excluded pre-RA patients (with CSA for example) from the study, thus skewing their results. CCP+ patients without synovitis are 50% likely to develop clinical RA in the subsequent 3 years, hence it is the best predictor of pre-clinical RA.

5. Figure 4 is backwards in the text compared to what is actually in the figure.

6. Line 163: the statement appears to be misworded (they do NOT return?) “Although levothyroxine brings down the level of cytokines , they do return to the same level as healthy controls after euthyroid state is achieved (18)

Again – this all gets back to the central question of what the study is actually trying to answer. Is this a study that “predicts clinical features of early RA” as stated in line 24 or correlate with those patients who have evidence of “rheumatism” as per line 248. This paper has to be rewritten in more clear, precise, and consistent terminology of what is being studied: I would recommend stating that lower (not necessarily low) levels of DHEAS in women with subclinical hypothyroidism correlate with clinically suspect arthralgia that is at higher risk for evolving into rheumatoid arthritis. I would also mention in the paper that the data do not support a correlation between DHEAS levels in general and higher risk arthralgias, although there is a weak but statistically significant association between high risk arthralgia and the lowest quartile of serum DHEAS levels, suggesting the possibility of a threshold effect of DHEAS at physiologic levels.

6. PLOS authors have the option to publish the peer review history of their article (what does this mean?). If published, this will include your full peer review and any attached files.

Reviewer #1: No

Reviewer #2: No

---

## [Author Response · Author response to Decision Letter 0]

24 Dec 2020

Reviever 1

Q1. Several abbreviations are used in the abstract without defining them: DHEAS, EULAR CSA, TPO, hsCRP.

While the meaning is clear and understandable, the manuscript should be edited for English grammar.

Response: We have added abbrevations and rectified grammatical errors

Q2. The use of hsCRP rather than CRP for the measurement of inflammation in the context of arthritis is not universally accepted, as the meaning of hsCRP in this context is not as well-documented as that of CRP or ESR

Response: We agree that CRP has been traditional and widely accepted marker of RA. hsCRP measures CRP in 5 to 10 mg/dl range . Its increased sensitivity may come at cost of specificity. Last few years have seen increase in use of hsCRP as disease activity marker (1) Since hsCRP is now widely available, most of our arthritis patients get done hsCRP. From perspective of our study, in which we excluded those with established RA, and we need to pick up inflammation (even at CRP <10) , use of hs CRP is more pragmatic.

1. Dessein PH, Joffe BI, Stanwix AE. High sensitivity C-reactive protein as a disease activity marker in rheumatoid arthritis. J Rheumatol. 2004 Jun;31(6):1095-7. PMID: 15170920.

Q3 The absence of males from the cohort significantly weakens the generalizability of the results.

Response: Both Rheumatoid arthritis and primary hypothyroidism is more common in females. In fact most studies of RA , including validation for EULAR has disproportionate female preponderance.

Q4 The sensitivity of DHEAS for predicting CSA was low, DHEAS levels < 43.6 while statistically significant did not have very strong predictive value of joint pains, and as seen in Figure 4, the confidence intervals of the mean DHEAS levels in those who did and did not meet CSA status were overlapping.

Response: There was overlapping as shown in figure 4 . That’s why, DHEAS levels , although low in CSA were not statistically significant. However, DHEAS < 43.6 had a specificity of 82 % and sensitivity of 52%, which is fairly acceptable for a retrospective analysis. If we prospectively do a study specifically asking and examining joint pains in all hypothyroid patients, the sensitivity will improve.

Q5 The statement in the discussion that DHEA supplementation has a proven role in SLE and PMR is not widely accepted, and the authors provide only one reference article to support this controversial statement in SLE and no evidence at all to support such a claim in PMR.

Response:

Thank you. I understand we should not be so emphatic. We have changed “DHEA supplementation has a proven role in SLE and PMR” to “ DHEA has been tried in SLE with some success (1) and has been suggested as therapeutic option in PMR (2)” We also change the reference accordingly 

Regarding SLE and DHEA : There have been 7 RCTs till 2007 of which only one found benefit. We have attached Cochrane reference in which authors concluded “there was evidence that DHEA had a modest but clinically significant impact on health related quality of life in the short term”

Regarding PMR and DHEA: 

We thank you for pointing out that there are no studies evaluating DHEA supplementation in PMR. The authors of one study (2)conclude “combination therapy with corticosteroids and DHEA may be a better therapeutic approach than prednisolone monotherapy.” 

1. https://www.cochranelibrary.com/cdsr/doi/10.1002/14651858.CD005114.pub2/full

2. R. H. Straub, T. Glück, M. Cutolo, J. Georgi, K. Helmke, J. Schölmerich, P. Vaith, B. Lang, The adrenal steroid status in relation to inflammatory cytokines (interleukin‐6 and tumour necrosis factor) in polymyalgia rheumatica, Rheumatology, Volume 39, Issue 6, June 2000, Pages 624–631, https://doi.org/10.1093/rheumatology/39.6.624

It came as surprise to us there are no studies of DHEA supplementation in PMR , especially since Low DHEAS is well documented in PMR. We quote a total of 13 studies. However , we do not wish them to add in references

1: Narváez J, Bernad B, Díaz Torné C, Momplet JV, Montpel JZ, Nolla JM,

Valverde-García J. Low serum levels of DHEAS in untreated polymyalgia

rheumatica/giant cell arteritis. J Rheumatol. 2006 Jul;33(7):1293-8. Epub 2006

Jun 15. PMID: 16783861.

2: Nilsson E, de la Torre B, Hedman M, Goobar J, Thörner A. Blood

dehydroepiandrosterone sulphate (DHEAS) levels in polymyalgia rheumatica/giant

cell arteritis and primary fibromyalgia. Clin Exp Rheumatol. 1994 Jul-

Aug;12(4):415-7. PMID: 7955606.

3: Cutolo M, Montecucco CM, Cavagna L, Caporali R, Capellino S, Montagna P,

Fazzuoli L, Villaggio B, Seriolo B, Sulli A. Serum cytokines and steroidal

hormones in polymyalgia rheumatica and elderly-onset rheumatoid arthritis. Ann

Rheum Dis. 2006 Nov;65(11):1438-43. doi: 10.1136/ard.2006.051979. Epub 2006 Apr

27. PMID: 16644782; PMCID: PMC1798362.

4: Cutolo M, Straub RH, Foppiani L, Prete C, Pulsatelli L, Sulli A, Boiardi L,

Macchioni P, Giusti M, Pizzorni C, Seriolo B, Salvarani C. Adrenal gland

hypofunction in active polymyalgia rheumatica. effect of glucocorticoid

treatment on adrenal hormones and interleukin 6. J Rheumatol. 2002

Apr;29(4):748-56. PMID: 11950017.

5: Straub RH, Glück T, Cutolo M, Georgi J, Helmke K, Schölmerich J, Vaith P,

Lang B. The adrenal steroid status in relation to inflammatory cytokines

(interleukin-6 and tumour necrosis factor) in polymyalgia rheumatica.

Rheumatology (Oxford). 2000 Jun;39(6):624-31. doi:

10.1093/rheumatology/39.6.624. PMID: 10888707.

6: Sulli A, Montecucco CM, Caporali R, Cavagna L, Montagna P, Capellino S,

Fazzuoli L, Seriolo B, Alessandro C, Secchi ME, Cutolo M. Glucocorticoid effects

on adrenal steroids and cytokine responsiveness in polymyalgia rheumatica and

elderly onset rheumatoid arthritis. Ann N Y Acad Sci. 2006 Jun;1069:307-14. doi:

10.1196/annals.1351.029. PMID: 16855158.

7: Pacheco MJ, Amado JA, Lopez-Hoyos M, Blanco R, Garcia-Unzueta MT, Rodriguez-

Valverde V, Martinez-Taboada VM. Hypothalamic-pituitary-adrenocortical axis

function in patients with polymyalgia rheumatica and giant cell arteritis. Semin

Arthritis Rheum. 2003 Feb;32(4):266-72. doi: 10.1053/sarh.2003.49993. PMID:

12621591.

8: Cutolo M, Foppiani L, Minuto F. Hypothalamic-pituitary-adrenal axis

impairment in the pathogenesis of rheumatoid arthritis and polymyalgia

rheumatica. J Endocrinol Invest. 2002;25(10 Suppl):19-23. PMID: 12508908.

9: Demir H, Tanriverdi F, Ozoğul N, Caliş M, Kirnap M, Durak AC, Keleştimur F.

Evaluation of the hypothalamic-pituitary-adrenal axis in untreated patients with

polymyalgia rheumatica and healthy controls. Scand J Rheumatol. 2006 May-

Jun;35(3):217-23. doi: 10.1080/03009740500474586. PMID: 16766369.

10: Pearce G, Ryan PF, Delmas PD, Tabensky DA, Seeman E. The deleterious effects

of low-dose corticosteroids on bone density in patients with polymyalgia

rheumatica. Br J Rheumatol. 1998 Mar;37(3):292-9. doi:

10.1093/rheumatology/37.3.292. PMID: 9566670.

11: Cutolo M, Sulli A, Pizzorni C, Craviotto C, Prete C, Foppiani L, Salvarani

C, Straub RH, Seriolo B. Cortisol, dehydroepiandrosterone sulfate, and

androstenedione levels in patients with polymyalgia rheumatica during twelve

months of glucocorticoid therapy. Ann N Y Acad Sci. 2002 Jun;966:91-6. doi:

10.1111/j.1749-6632.2002.tb04206.x. PMID: 12114263.

12: Cutolo M, Sulli A, Pizzorni C, Craviotto C, Straub RH. Hypothalamic-

pituitary-adrenocortical and gonadal functions in rheumatoid arthritis. Ann N Y

Acad Sci. 2003 May;992:107-17. doi: 10.1111/j.1749-6632.2003.tb03142.x. PMID:

12794051.

13: de la Torre B, Fransson J, Scheynius A. Blood dehydroepiandrosterone

sulphate (DHEAS) levels in pemphigoid/pemphigus and psoriasis. Clin Exp

Rheumatol. 1995 May-Jun;13(3):345-8. PMID: 7554562.

Q6 Overall, the paper's conclusion that the potential role of DHEAS in improving low QoL in hypothyroid patients with joint pain and normal TSH should be further investigated is not strongly supported by the findings described.

Response: Thank you for pointing this out. We have changed “the potential role of DHEAS in improving low QoL in hypothyroid patients with joint pain and normal TSH should be further investigated” to “the potential role of DHEAS supplementation in natural history of hypothyroid patients with joint pains”

Reviever 2

To start, the terminology, methods, and hypothesis used in this study are confusing. Is this a study to demonstrate a correlation of DHEAS levels with “early RA,” pre-RA, or simply with “rheumatism” - which I take to mean joint pains and arthralgia but not clinical synovitis (the term “rheumatism” is not precisely defined by the authors). This is not a trivial point. In fact, it is central to the whole narrative of the study. Yet, all of these terms are used interchangeably throughout the paper.

To be clear, “early RA” is not “pre-RA”, and rheumatism can describe any, all, or neither of those terms.

The EULAR-CSA score was established to identify specific patients in the pre-clinical stage of RA disease development, as RA likely develops in a multi-step approach from pre-clinical autoimmunity to overt systemic and articular inflammatory disease. The score does not determine who has early (classifiable) RA. Indeed, the EULAR CSA score was validated against clinical acumen that was able at its best to identify a 20% chance of someone with CSA developing early RA in the next few years.

Response: By “rheumatism” we mean “clinical features of pre RA as mentioned in EULAR CSA score “ and not early RA. EULAR CSA criteria is for pre RA, but in the entire document they have used the word “imminent RA”. Thus imminent RA or high risk of RA and pre RA convey same meaning.

 In line 200 we have used the term “pre-RA like” not pre-RA. As shown below the scope of conditions qualifing for pre-RA has been expanding. By usage of “pre-RA like” we speculate {lower DHEAS + primary hypothyroidism} subjects may fit into some of following pre-RA criteria (ref). Our present study is to have basis for further research in this direction. 

Pre-RA

a. Genetic risk factors for RA 

b. Environmental risk factors for RA Asymptomatic

c. Systemic autoimmunity associated with RA 

d. Symptoms without clinical evidence of arthritis 

e. Early undifferentiated arthritis. Symptomatic

Ref: 

Gerlag DM, Raza K, Van Baarsen LG, Brouwer E, Buckley CD, Burmester GR, et al. EULAR recommendations for terminology and research in individuals at risk of rheumatoid arthritis: report from the Study Group for Risk Factors for Rheumatoid Arthritis. Ann Rheum Dis. 2012;71:638–41.

1. DHEAS levels significantly and inversely correlated with CSA score. However – they did NOT correlate with CSA being present. When the CSA score was dichotomized, binary logistic regression found no association (an odds ratio of 1.01) between whether CSA was present or absent and DHEAS levels. And mean DHEAS was not significantly different in those with CSA and those without. So the statement that “serum DHEAS predicted rheumatism” is NOT correct (line 109) based on those data. Again, it depends upon how one defines “rheumatism,” because MCP pain can be considered a feature of “rheumatism,” but DHEA as a whole did not correlate with higher risk arthralgias predictive of developing RA as defined by a CSA score>4.

Response : When binary logistic regression was done it was done for particular EULAR criteria (seven in all) The OR of 1.01 assuaged the symmetry of EULAR i.e no one criteria (for example morning stiffness) was driving the significance . The mean DHEAS level in CSA absent and CSA present was not statistically significant. This does not mean DHEA does not correlare. In results section we have given the correlation analysis (line 129)

2. When DHEAS levels were dichotomized as well, then there was a correlation between a DHEAS level <43.6 and CSA present. The author’s state that “low levels of DHEAS” predict clinical features of “early RA” in primary hypothyoid patients. This is misleading. First, is DHEAS < 43.6 low or not? Physiologic rather than pathological levels of DHEA may be relevant to the immunologic/inflammatory effects mentioned in the paper. No reference is made to what is considered a normal level physiologically in women other than general mention of lower and higher levels by age and menopausal status. Second, the authors selectively drew their cutoff at that level not because it was low physiologically per se but because it granted them the most favorable characteristics on an ROC curve, such that only those DHEAS levels in the bottom quartile were considered. And even then, the AUC was weak. 

Response: We have not deliberately mentioned level considered physiological in women, coz while normative ranges of DHEAS levels in healthy population are available, its not known what are normal levels in any of disease states..We agree with your suggestion that instead of “lower” we should use the term “low” . 

The AUC is 0.67 (line 114) which is reasonably strong for clinical interpretation. 

Third, because DHEAS levels correlate with age and other factors (as shown, albeit weakly) selectively reporting results comparing only the lowest quartile could introduce bias by comparing patients with different demographic characteristics (and lower DHEAS levels) to those with higher levels and different demographics.

Response:

As linear correlation age and DHEAS were inversely correlated. This fact we have mentioned in results. We are also attaching Spearman and Pearson correlation results for reference

Coefficientsa

Model Unstandardized Coefficients Standardized Coefficients t Sig. 95.0% Confidence Interval for B

 B Std. Error Beta Lower Bound Upper Bound

1 (Constant) 4.833 .290 16.643 .000 4.255 5.412

 DHEAS -.004 .002 -.224 -2.002 .049 -.009 .000

2 (Constant) 4.880 .950 5.136 .000 2.987 6.773

 DHEAS -.004 .002 -.225 -1.943 .056 -.009 .000

 Age -.001 .020 -.006 -.052 .959 -.040 .038

a. Dependent Variable: EULARscore

Regardiing the effect of age on quaratiles (which could potentially affect the results), we did one way ANOVA for age in four quartile of DHEAS and it was statistically non significant. We are attaching table for reference.

ANOVA

Age 

 Sum of Squares df Mean Square F Sig.

Between Groups 702.515 3 234.172 2.364 .078

Within Groups 7331.332 74 99.072 

Total 8033.846 77 

Other issues:

3. Were the assessors of CSA blinded to DHEAS levels? I assume since CSA is part of the clinical practice and that this is a retrospective review, the answer is that the assessors had no knowledge of present or past DHEAS levels. But that needs to be stated.

Assesors of CSA were blind to DHEAS levels. We have stated that in revised version (Line 79-81)

4. How were patients with RA excluded? Was this based on ACR 1987 or ACR/EULAR 2010 criteria? Or simply a “clinical diagnosis” of RA. If the latter, then this is problematic. The authors did not measure CCP levels in their patients, but CCP+ patients (even those without arthritis) could have a diagnosis of “RA” in their chart because of the positive test and thus could have been excluded. If so, then the authors may have excluded pre-RA patients (with CSA for example) from the study, thus skewing their results. CCP+ patients without synovitis are 50% likely to develop clinical RA in the subsequent 3 years, hence it is the best predictor of pre-clinical RA.

Response: ACR 1987 criteria was used to exclude RA. In addotion those who had documentation of RA diagnosis in past with /without DMARDs were also deemed to have RA and excluded .Also a previous history of RA and /or being on DMARD. We understand the drawback of 1987 criteria .We did not have anti-CCP and this we have acknowledged as well. Due to retrospective nature of study, we agree we may have missed pre RA patients. While there are a number of studies of hypothyroidism in RA,there are no studies on prevalence of anti-CCP in hypothyrodism. There is no historical cohort or published literature to “estimate” how many anti-CCP + patients without synovitis hypothyroidism we might have missed. This is selection bias which is inherent in all observational studies.

5. Figure 4 is backwards in the text compared to what is actually in the figure

Figure 4 is now mentioned in line no 133, after figures 1 , 2 and 3 in Results section

6. Line 163: the statement appears to be misworded (they do NOT return?) “Although levothyroxine brings down the level of cytokines , they do return to the same level as healthy controls after euthyroid state is achieved (18)

Response: Thank you. We have rectified typo error

Again – this all gets back to the central question of what the study is actually trying to answer. Is this a study that “predicts clinical features of early RA” as stated in line 24 or correlate with those patients who have evidence of “rheumatism” as per line 248. This paper has to be rewritten in more clear, precise, and consistent terminology of what is being studied: I would recommend stating that lower (not necessarily low) levels of DHEAS in women with subclinical hypothyroidism correlate with clinically suspect arthralgia that is at higher risk for evolving into rheumatoid arthritis. 

Thank you for valuable suggestion. We have revised to incorporate “lower (not low) levels of DHEAS” in line 26 and line 31. Actually this is what we wanted to emphasize.We accept in line 248 (which is conclusion), we should not use vague terminology like “rheumatism”. We have corrected “rheumatism” for “clinical features of early RA”. To make the maniuscipt more precise we

I would also mention in the paper that the data do not support a correlation between DHEAS levels in general and higher risk arthralgias, although there is a weak but statistically significant association between high risk arthralgia and the lowest quartile of serum DHEAS levels, suggesting the possibility of a threshold effect of DHEAS at physiologic level

Yes we agree the data does not support correlation between DHEAS and risk of arthralgias in linear manner. But there is correlation with DHEAS quartiles.As shown, lower DHEAS levels do correlate. 

We have hypothesized a mechanism for threshold effect of DHEAS at physiologic level in discussion section, including the possibility of further studies, preferably prospective to delineate natural history.

---

## [Editor Report · Decision Letter 1]

15 Jan 2021

Dehydroepiandrostenedione Sulphate  (DHEAS)  Levels Predict high risk of  Rheumatoid Arthritis (RA) in Subclinical Hypothyroidism

PONE-D-20-22641R1

Dear Dr. 

We’re pleased to inform you that your manuscript has been judged scientifically suitable for publication and will be formally accepted for publication once it meets all outstanding technical requirements.

Kind regards,

Rosanna Di Paola, MD

Academic Editor

PLOS ONE
---

## [Editor Report · Acceptance letter]

25 Jan 2021

PONE-D-20-22641R1 

Dehydroepiandrostenedione Sulphate  (DHEAS)  Levels Predict high risk of  Rheumatoid Arthritis (RA) in Subclinical Hypothyroidism 

Dear Dr. shukla:

I'm pleased to inform you that your manuscript has been deemed suitable for publication in PLOS ONE. Congratulations! Your manuscript is now with our production department. 

Kind regards, 

on behalf of

Dr. Rosanna Di Paola 

Academic Editor

PLOS ONE